OBSERVATION
Clinical Science and Epidemiology

# Handwashing and Detergent Treatment Greatly Reduce SARS-CoV-2 Viral Load on Halloween Candy Handled by COVID-19 Patients

Rodolfo A. Salido,[a] Sydney C. Morgan,[b] Maria I. Rojas,[c,m] Celestine G. Magallanes,[b] Clarisse Marotz,[d] Peter DeHoff,[b] Pedro Belda-Ferre,[d] Stefan Aigner,[e,f,g] Deborah M. Kado,[h,i] Gene W. Yeo,[e,f,g] Jack A. Gilbert,[d,j,k] Louise Laurent,[b] Forest Rohwer,[c,m] Rob Knight[a,d,j,k,l]

[a]Department of Bioengineering, University of California, San Diego, La Jolla, California, USA

[b]Department of Obstetrics, Gynecology, and Reproductive Science, University of California, San Diego, La Jolla, California, USA

[c]Department of Biology, San Diego State University, San Diego, California, USA

[d]Department of Pediatrics, University of California, San Diego, La Jolla, California, USA

[e]Department of Cellular and Molecular Medicine, University of California, San Diego, La Jolla, California, USA

[f]Stem Cell Program, University of California, San Diego, La Jolla, California, USA

[g]Institute for Genomic Medicine, University of California, San Diego, La Jolla, California, USA

[h]Herbert Wertheim School of Public Health and Human Longevity Science, University of California, San Diego, La Jolla, California, USA

[i]Department of Medicine, University of California, San Diego, La Jolla, California, USA

[j]Scripps Institution of Oceanography, University of California, San Diego, La Jolla, California, USA

[k]Center for Microbiome Innovation, University of California, San Diego, La Jolla, California, USA

[l]Department of Computer Science and Engineering, University of California, San Diego, La Jolla, California, USA

[m]Viral Information Institute, San Diego State University, San Diego, California, USA

Rodolfo A. Salido, Sydney C. Morgan, and Maria I. Rojas contributed equally to this work. Author order was determined by consensus among the corresponding authors based on a written description of the contribution of each author.

**ABSTRACT** Although SARS-CoV-2 is primarily transmitted by respiratory droplets and aerosols, transmission by fomites remains plausible. During Halloween, a major event for children in numerous countries, SARS-CoV-2 transmission risk via candy fomites worries many parents. To address this concern, we enrolled 10 recently diagnosed asymptomatic or mildly/moderately symptomatic COVID-19 patients to handle typical Halloween candy (pieces individually wrapped) under three conditions: normal handling with unwashed hands, deliberate coughing and extensive touching, and normal handling following handwashing. We then used a factorial design to subject the candies to two posthandling treatments: no washing (untreated) and household dishwashing detergent. We measured SARS-CoV-2 load by reverse transcriptase quantitative PCR (RT-qPCR) and loop-mediated isothermal amplification (LAMP). From the candies not washed posthandling, we detected SARS-CoV-2 on 60% of candies that were deliberately coughed on, 60% of candies normally handled with unwashed hands, but only 10% of candies handled after hand washing. We found that treating candy with dishwashing detergent reduced SARS-CoV-2 load by 62.1% in comparison to untreated candy. Taken together, these results suggest that although the risk of transmission of SARS-CoV-2 by fomites is low even from known COVID-19 patients, viral RNA load can be reduced to near zero by the combination of handwashing by the infected patient and ≥1 min detergent treatment after collection. We also found that the inexpensive and fast LAMP protocol was more than 80% concordant with RT-qPCR.

**IMPORTANCE** The COVID-19 pandemic is leading to important tradeoffs between risk of severe acute respiratory syndrome coronavirus 2 (SARS-CoV-2) transmission and mental health due to deprivation from normal activities, with these impacts be-

Address correspondence to Louise Laurent, llaurent@ucsd.edu, Forest Rohwer, frohwer@gmail.com, or Rob Knight, robknight@ucsd.edu.

ing especially profound in children. Due to the ongoing pandemic, Halloween activities will be curtailed as a result of the concern that candy from strangers might act as fomites. Here, we demonstrate that these risks can be mitigated by ensuring that, prior to handling candy, the candy giver washes their hands and, after receipt, by washing candy with household dishwashing detergent. Even in the most extreme case, with candy deliberately coughed on by known COVID-19 patients, viral load was reduced dramatically after washing with household detergent. We conclude that with reasonable precautions, even if followed only by either the candy giver or the candy recipient, the risk of viral transmission by this route is very low.

**KEYWORDS** COVID-19, Halloween, LAMP, RT-qPCR, SARS-CoV-2, candy, fomite, qPCR, surface swab

The COVID-19 pandemic has caused >8 million cases and >220,000 deaths in the United States alone as of mid-October 2020. Fear of infection has severely curtailed normal activities. In the United States, Halloween is a major children's holiday, but fear of SARS-CoV-2 transmission by candy is leading many parents to plan on keeping their children from participating. Quantifying the risk of SARS-CoV-2 transmission by candy is therefore of compelling interest.

Accordingly, we enrolled 10 recently diagnosed COVID-19 positive outpatients into an Institutional Review Board (IRB)-approved study, confirming these individuals as positive on the day of testing using an anterior nares clinical reverse transcriptase quantitative PCR (RT-qPCR) assay in the UC San Diego (UCSD) EXCITE (EXpedited COVID-19 IdenTification Environment) laboratory. Patients were recruited to the study via phone call after they tested positive via UCSD Health, were aged 18 to 55 (7 female and 3 male), and were either mildly/moderately symptomatic (9/10) or asymptomatic (1/10). Six of the symptomatic patients reported symptoms of cough. Each individual was provided with four bags containing two Halloween candies (combination of Haribo gummies, Twix, M&Ms, Starburst, and Snickers), allocated as follows: bag 1, candies handled normally with unwashed hands; bag 2, candies coughed on and handled with unwashed hands; bag 3, candies handled after thorough hand washing with soap for 20 s, per CDC guidelines; and bag 4, left as a thank you gift to the subject for their participation. In a factorial design, we then assigned one candy from each bag to either untreated/no washing, or household dishwashing detergent containing sodium lauryl sulfate (SLS) diluted 1:50, for ≥1 min (Fig. S1 in the supplemental material).

RNA was extracted from swabs used to sample the surface of the candy wrappers with the Omega MagBind Viral DNA/RNA kit on the Kingfisher platform, followed by RT-qPCR using the Thermo TaqPath COVID-19 Multiplex assay on the Quantstudio 7 Pro platform. Loop-mediated isothermal amplification (LAMP) was performed in multiplex using NEB WarmStart Colorimetric LAMP Master Mix with UDG (1–3).

Surprisingly, coughing on and extensive handling of candies yielded the same SARS-CoV-2 positivity rate as unwashed handling (5 of 10 untreated candies, 6 of 30 total). Hand washing prior to handling greatly reduced positives (1 of 10 untreated candies, 2 of 30 total) (Fig. 1A). A detergent posthandling treatment reduced the viral load significantly, with average quantification cycle ($Cq$) of 33.12 ± 1.78 for untreated samples ($n = 29$), increasing to 35.70 ± 1.50 for detergent-treated candy ($n = 9$). Mean $Cq$ for untreated candy was significantly different from detergent-treated candy ($t = -3.93$, $P = 0.000372$) using a two-tailed independent sample $t$ test (Fig. 2A). Our subjects had mild/moderate cases of COVID-19 with possibly low viral load: subject SB_0154 may have been an outlier with high load, contributing to detection of SARS-CoV-2 on candy from that individual even after handwashing and detergent treatment.

An important limitation in dealing with environmental samples is the usual unavailability of high-capacity qPCR machines. We found good concordance between the

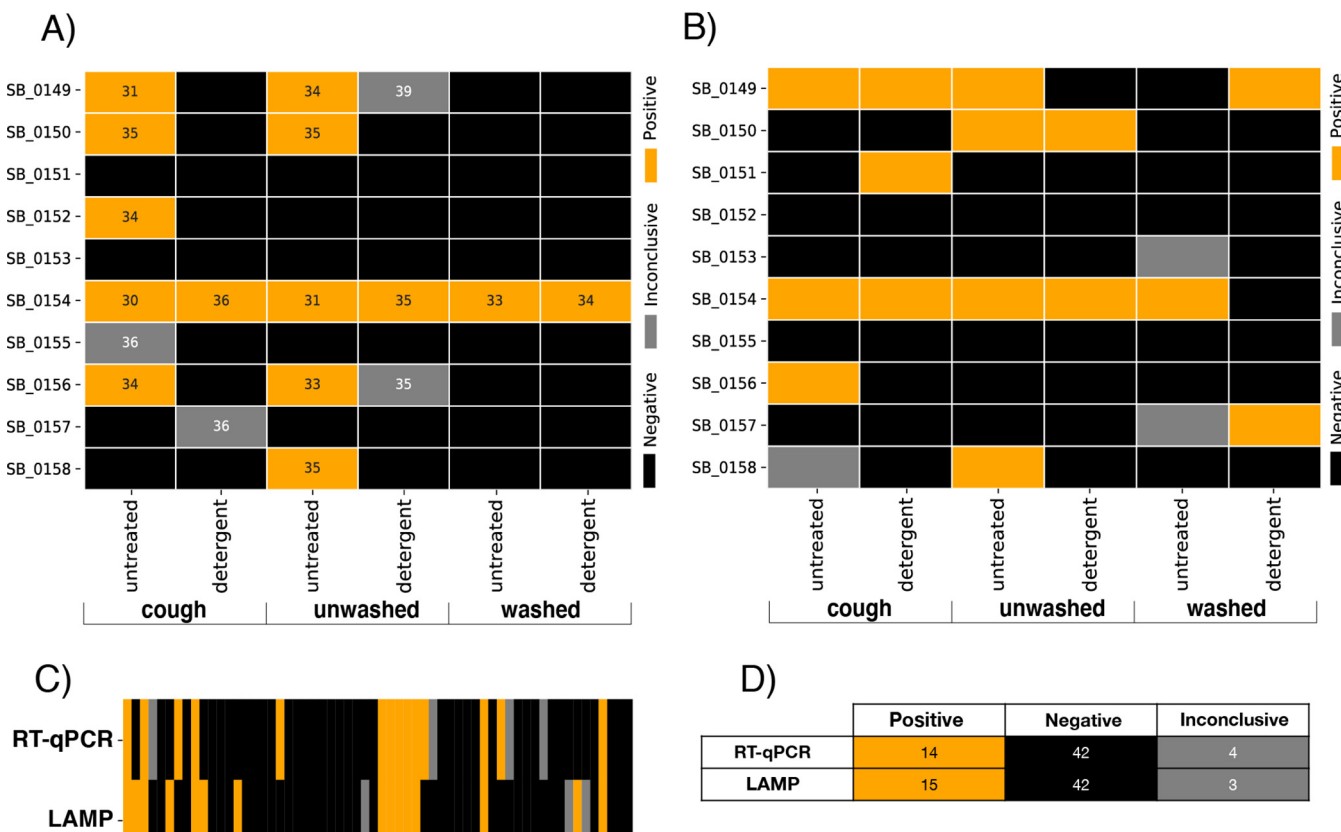

**FIG 1** (A) Heatmap of results from SARS-CoV-2 RT-qPCR detection assay with subjects as rows and treatment groups as nested columns. Positive results were called on samples for which at least two out of three viral genes were detected. Inconclusive results were called when only one viral gene was detected and negative results when no viral genes were detected. Positive results display the average $Cq$ values across detected genes, while inconclusive results report a single $Cq$ value for the detected gene. (B) Heatmap of results from SARS-CoV-2 LAMP detection assay. Positive results were called on colorimetric readouts that matched the positive control, while negative results were called on colorimetric matches to the negative control. Inconclusive results were called on RT-PCRs that were inhibited. (C) Heatmap of comparison of results between RT-qPCR and LAMP. The heatmap shows 83% concordance when excluding the negative to inconclusive mismatches. (D) Classification matrix comparing SARS-CoV-2 detection results between RT-qPCR and LAMP.

LAMP assay and RT-qPCR, with 44 of 53 samples concordant between the two assays (83% concordance), excluding inconclusive test results. Result outcomes did not differ significantly between RT-qPCR and LAMP ($\chi^2 = 0.399$, df = 2, $P = 0.818$) (Fig. 1D). Therefore, LAMP is an effective screen for environmental samples, although RT-qPCR allows quantitative measurements.

Overall, these results have the following implications. First, even candies handled by a known COVID-19 patient have low or undetectable viral loads if reasonable precautions, such as hand washing before handling candies, or washing candies with detergent after collecting, are employed. Second, even candies that have been deliberately contaminated by coughing can be treated with mild household dishwashing detergent, such that only 20% have any detectable signal by RT-qPCR. A treatment of ≥1 min with SLS from domestic dishwashing detergent can reduce infectious viral particles by 3,000-fold (4–6). An important limitation of this experiment is that we only studied viral RNA, not infectious particles; detection of SARS-CoV-2 viral RNA does not necessarily indicate infectious virus. Previous work reported that only samples with SARS-CoV-2 detected with $Cq < 30$ were capable of infection (7).

Although transmission of SARS-CoV-2 by fomites is thought to be low (8), reasonable precautions both in handing out candy and receiving it should make fomite transmission risk negligible. Because the primary risk at Halloween is droplet or

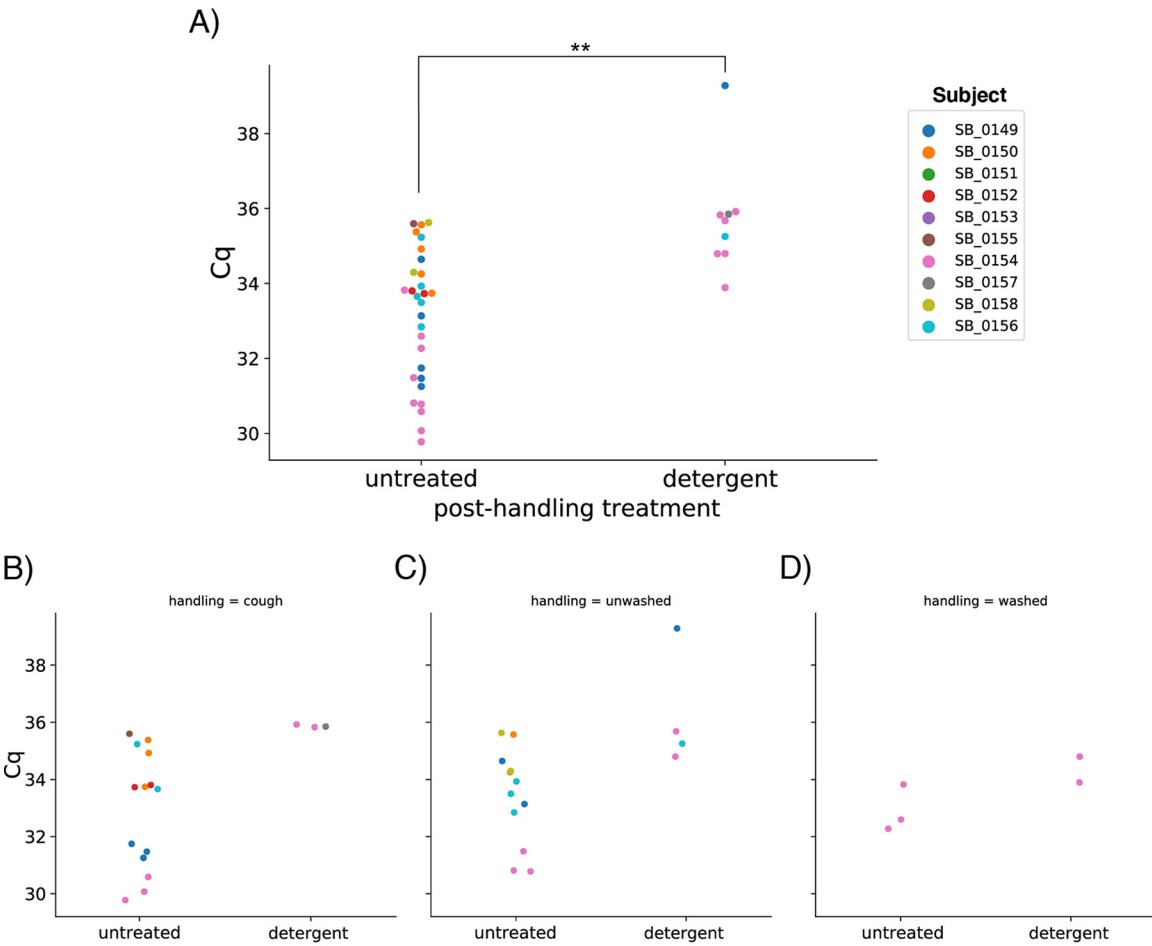

**FIG 2** (A) Swarmplot of *Cq* values for detected viral genes across all candy grouped into posthandling treatments. The distribution of *Cq* values for the untreated candy was significantly different than that of detergent (\*\*, *P* < 0.001). (B to D) Swarmplots of *Cq* values for detected viral genes divided by handling conditions and grouped into posthandling treatments. Both the extensively handled and coughed-on candy (B) and the candy that was normally handled with unwashed hands (C) had detectable viral genes with comparable *Cq* values ranging from 29.77 to 39.28. Treating with detergent reduced the viral load on candies, measured as an increase in *Cq*, or resulted in undetectable viral load. Washing hands before handling candy (D) markedly decreased viral gene detection rate and decreased viral load on positive candies.

airborne transmission from other people, the CDC recommends social distancing, hand washing, distanced pickup of candy, and, of course, masks.

## SUPPLEMENTAL MATERIAL

Supplemental material is available online only.

**TEXT S1**, DOCX file, 0.02 MB.

**FIG S1**, TIF file, 0.2 MB.

**TABLE S1**, DOCX file, 0.01 MB.

**TABLE S2**, DOCX file, 0.01 MB.

**TABLE S3**, DOCX file, 0.01 MB.

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
