## [Reviewer comments · mSystems]

Handwashing and detergent treatment greatly reduce SARS-CoV-2 viral load on Halloween candy handled by COVID-19 patients.

Rodolfo Salido, Sydney Morgan, Maria Rojas, Celestine Magallanes, Clarisse Marotz, Peter DeHoff, Pedro Belda-Ferre, Stefan Aigner, Deborah Kado, Gene Yeo, Jack Gilbert, Louise Laurent, Forest Rohwer, and Rob Knight

Corresponding Author(s): Rob Knight, UCSD School of Medicine

Review Timeline:

Submission Date:	October 16, 2020
Editorial Decision:	October 28, 2020
Revision Received:	October 29, 2020
Accepted:	October 29, 2020

Editor: Paola de Sessions

Reviewer(s): The reviewers have opted to remain anonymous.

Transaction Report:

DOI: <https://doi.org/10.1128/mSystems.01074-20>

October 28, 2020

Dr. Rob Knight
University of California, San Diego
La Jolla

Re: mSystems01074-20 (Handwashing and detergent treatment greatly reduce SARS-CoV-2 viral load on Halloween candy handled by COVID-19 patients.)

Dear Dr. Rob Knight:

Reviewer 1 mentioned "Manuscript could be improved with a schematic of the experimental design. I worry about the implications of using bleach to show disinfection, saying that it is more effective than detergent, and then saying do not use it. It seems like mixed messaging and irresponsible from a public health standpoint (better not to present the hypochlorite data). I think that mentioning hypochlorous as an approved disinfectant further mixes the message (considering the disinfection chemistry)." I think is the biggest concern here, can this be clarified?

Below you will find the comments of the reviewers.

To submit your modified manuscript, log onto the eJP submission site at <https://msystems.msubmit.net/cgi-bin/main.plex>. If you cannot remember your password, click the "Can't remember your password?" link and follow the instructions on the screen. Go to Author Tasks and click the appropriate manuscript title to begin the resubmission process. The information that you entered when you first submitted the paper will be displayed. Please update the information as necessary. Provide (1) point-by-point responses to the issues raised by the reviewers as file type "Response to Reviewers," not in your cover letter, and (2) a PDF file that indicates the changes from the original submission (by highlighting or underlining the changes) as file type "Marked Up Manuscript - For Review Only."

Due to the SARS-CoV-2 pandemic, our typical 60 day deadline for revisions will not be applied. I hope that you will be able to submit a revised manuscript soon, but want to reassure you that the journal will be flexible in terms of timing, particularly if experimental revisions are needed. When you are ready to resubmit, please know that our staff and Editors are working remotely and handling submissions without delay. If you do not wish to modify the manuscript and prefer to submit it to another journal, please notify me of your decision immediately so that the manuscript may be formally withdrawn from consideration by mSystems.

Sincerely,

Paola de Sessions

Editor, mSystems

Journals Department
Reviewer comments:

Reviewer #1 (Comments for the Author):

This is a fun and interesting study. However there are some considerable shortcomings in the study design. Further the conclusions drawn from the study are not fully supported.

General Comments:

The strengths of the study are the comparison of the RT-PCR to RT-LAMP, and the observation that candies can (albeit at relatively low levels) have detectable levels of viral RNA on them as a result of handling or being coughed on.

The number of samples were relatively limited and the study design was not appropriate to provide a rigorous evaluation the disinfection by washing with detergent or bleach. First, the methods use both target detection of RNA rather than viable virus. Second, the starting levels prior to treatment are highly variable due to the natural contamination route. As a result the sample number is too low to understand level of inactivation.

Specific comments:

Manuscript could be improved with a schematic of the experimental design.

I worry about the implications of using bleach to show disinfection, saying that it is more effective than detergent, and then saying do not use it. It seems like mixed messaging and irresponsible from a public health standpoint (better not to present the hypochlorite data). I think that mentioning hypochlorous as an approved disinfectant further mixes the message (considering the disinfection chemistry).

The use of the term disinfection has specific meaning (refer to EPA's OPP antimicrobial pesticide testing). The levels of reduction reported are not disinfection (in the technical sense).

The reliance on $C_q < 30$ as an indication of infectivity is an irresponsible extrapolation of the cited reference.

Risk of transmission is not adequately assessed as part of the methodology of this study. All that is considered is the presence of detectable virus.

I see nothing in the study that supports the CDC recommendation of placing candy in pre-prepared bags.

Reviewer #2 (Comments for the Author):

In this paper, the authors investigated the of SARS-CoV-2 transmission by Halloween candy via fomites. They found that although the risk of transmission of SARS-44 CoV-2 by fomites is low even from known COVID-19 patients, it can be reduced to near zero by the combination of handwashing by the infected patient and {greater than or equal to}1 minute detergent treatment after collection.

They detected SARS-CoV-2 on 60% of candies that were deliberately coughed on, 60% of candies normally handled with unwashed hands, but only 10% of candies handled after hand washing.

Additionally, they found that bleach was more effective than detergent at disinfecting candies, with bleach reducing SARS-CoV-2 load by 75.7%, relative to 62.1% for detergent. However, bleach leaked through some of the candy wrappers, making it unsafe to use in practice.

Lastly, they found good concordance between the LAMP assay and RT-qPCR, with 67 of 78 samples concordant between the two assays (85.8% concordance), excluding Inconclusive test results. Suggesting that LAMP can be an effective screen for environmental samples.

The study is well conducted and the discussion sound. In light of the relevance of the topic, the paper will be a timely addition to the literature.

Major corrections:

None

Minor corrections:

A discussion could be added to note that these are mild cases of SARS-CoV-2 infections. As such, the viral load of these individuals may be low. This may be the cause for the outlier subject SB_0154 (high viral load), whose samples were found to be contaminated despite handwashing prior or disinfection after.

Dear Dr. de Sessions,

Thanks so much for taking the time to handle our manuscript. In summary, we have removed the bleach data, added the schematic as a supplementary figure, and taken into account all the reviewers' comments and yours as described in detail below.

Best,
Rob

Dear Dr. Rob Knight:

Reviewer 1 mentioned "Manuscript could be improved with a schematic of the experimental design.

We have added this as Supplementary Figure S1.

I worry about the implications of using bleach to show disinfection, saying that it is more effective than detergent, and then saying do not use it. It seems like mixed messaging and irresponsible from a public health standpoint (better not to present the hypochlorite data). I think that mentioning hypochlorous as an approved disinfectant further mixes the message (considering the disinfection chemistry)." I think is the biggest concern here, can this be clarified?

We have removed all the bleach data and reference to hypochlorite as requested.

Below you will find the comments of the reviewers.

To submit your modified manuscript, log onto the eJP submission site at <https://msystems.msubmit.net/cgi-bin/main.plex>. If you cannot remember your password, click the "Can't remember your password?" link and follow the instructions on the screen. Go to Author Tasks and click the appropriate manuscript title to begin the resubmission process. The information that you entered when you first submitted the paper will be displayed. Please update the information as necessary. Provide (1) point-by-point responses to the issues raised by the reviewers as file type "Response to Reviewers," not in your cover letter, and (2) a PDF file that indicates the changes from the original submission (by highlighting or underlining the changes) as file type "Marked Up Manuscript - For Review Only."

Due to the SARS-CoV-2 pandemic, our typical 60 day deadline for revisions will not be applied. I hope that you will be able to submit a revised manuscript soon, but want to reassure you that the journal will be flexible in terms of timing, particularly if experimental revisions are needed. When you are ready to resubmit, please know that our staff and Editors are working remotely and handling submissions without delay. If you do not wish to modify the manuscript and prefer to submit it to another journal, please notify me of your decision immediately so that the manuscript may be formally withdrawn from consideration by mSystems.

Sincerely,

Paola de Sessions

Editor, mSystems

Journals Department
Reviewer comments:

Reviewer #1 (Comments for the Author):

This is a fun and interesting study. However there are some considerable shortcomings in the study design. Further the conclusions drawn from the study are not fully supported.

General Comments:

The strengths of the study are the comparison of the RT-PCR to RT-LAMP, and the observation that candies can (albeit at relatively low levels) be have detectable levels of viral RNA on them as a result of handling or being coughed on.

The number of samples were relatively limited and the study design was not appropriate to provide a rigorous evaluation the disinfection by washing with detergent or bleach. First, the methods use both target detection of RNA rather than viable virus. Second, the starting levels prior to treatment are highly variable due to the natural contamination route. As a result the sample number is too low to understand level of inactivation.

We note that in the United States, any experiments to measure viable viral load must be performed in a BSL3 containment facility. Access to BSL3 facilities is extremely limited, and we were not able to gain access to the facility at UCSD in a timely manner.. We do note that the level of reduction of viral RNA is measurable with this sample size. We agree that the comparison between detergent and bleach is problematic, and have removed the latter.

Specific comments:

Manuscript could be improved with a schematic of the experimental design.

We have added this as Fig. S1.

I worry about the implications of using bleach to show disinfection, saying that it is more effective than detergent, and then saying do not use it. It seems like mixed messaging and irresponsible from a public health standpoint (better not to present the hypochlorite data). I think that mentioning hypochlorous as an approved disinfectant further mixes the message (considering the disinfection chemistry).

We have removed all mention of bleach, and thank the reviewer for the suggestion.

The use of the term disinfection has specific meaning (refer to EPA's OPP antimicrobial pesticide testing). The levels of reduction reported are not disinfection (in the technical sense).

We did not intend "disinfection" in the technical sense, and have replaced it throughout with "reduction in viral RNA" or similar phrasing.

The reliance on Cq<30 as an indication of infectivity is an irresponsible extrapolation of the cited reference. Risk of transmission is not adequately assessed as part of the methodology of this study. All that is considered is the presence of detectable virus.

We have removed this interpretation of that study as requested.

I see nothing in the study that supports the CDC recommendation of placing candy in pre-prepared bags.

We apologize for the lack of clarity, and have re-worded the last sentence to highlight the CDC recommendations (wear a mask, social distance, pick up candy remotely) rather than to suggest that this specific study provides supporting evidence for all aspects of these recommendations.

Reviewer #2 (Comments for the Author):

In this paper, the authors investigated the of SARS-CoV-2 transmission by Halloween candy via fomites. They found that although the risk of transmission of SARS-44 CoV-2 by fomites is low even from known COVID-19 patients, it can be reduced to near zero by the combination of handwashing by the infected patient and {greater than or equal to}1 minute detergent treatment after collection.

They detected SARS-CoV-2 on 60% of candies that were deliberately coughed on, 60% of candies normally handled with unwashed hands, but only 10% of candies handled after hand washing. Additionally, they found that bleach was more effective than detergent at disinfecting candies, with bleach reducing SARS-CoV-2 load by 75.7%, relative to 62.1% for detergent. However, bleach leaked through some of the candy wrappers, making it unsafe to use in practice.

Lastly, they found good concordance between the LAMP assay and RT-qPCR, with 67 of 78 samples concordant between the two assays (85.8% concordance), excluding Inconclusive test results. Suggesting that LAMP can be an effective screen for environmental samples.

The study is well conducted and the discussion sound. In light of the relevance of the topic, the paper will be a timely addition to the literature.

We thank the reviewer for the kind comments.

Major corrections:

None

Minor corrections:

A discussion could be added to note that these are mild cases of SARS-CoV-2 infections. As such, the viral load of these individuals may be low. This may be the cause for the outlier subject SB_0154 (high viral load), whose samples were found to be contaminated despite handwashing prior or disinfection after.

We have added these points briefly to the discussion, and agree with the reviewer's interpretation.

October 29, 2020

Prof. Rob Knight
UCSD School of Medicine
9500 Gilman Drive
MC 0602
La Jolla, CA 92093

Re: mSystems01074-20R1 (Handwashing and detergent treatment greatly reduce SARS-CoV-2 viral load on Halloween candy handled by COVID-19 patients.)

Dear Prof. Rob Knight:

This is a very relevant manuscript and extremely timely in a period when science has come into question. It has the potential to reach anyone interested in public health and holidays!

Your manuscript has been accepted, and I am forwarding it to the ASM Journals Department for publication. For your reference, ASM Journals' address is given below. Before it can be scheduled for publication, your manuscript will be checked by the mSystems senior production editor, Ellie Ghatineh, to make sure that all elements meet the technical requirements for publication. She will contact you if anything needs to be revised before copyediting and production can begin. Otherwise, you will be notified when your proofs are ready to be viewed.

Sincerely,

Paola de Sessions
Editor, mSystems

Journals Department
Supplemental Material: Accept
Supplemental Material: Accept